# Revealing Halos Concealed by Cirrus Clouds

Yuji Ayatsuka[1]

[1]Personal Project, Tokyo, Japan

**Correspondence:** Yuji Ayatsuka (ayatsuka@acm.org)

**Abstract.** Many types of halos (including arcs) appear in the sky. Each type of halo corresponds to the shape and orientation of ice crystals in clouds, and reflects the state of the atmosphere, therefore observing them from the ground greatly helps in understanding the state of the atmosphere. However, halos are easily obscured by the contrast of the cloud itself, making it difficult to observe them. This difficulty can be overcome by enhancing halos on images, for which various techniques have been developed. This study describes the construction of a sky-color model for halos and a new effective algorithm to reveal halos on images.

## 1 Introduction

Ice crystals that form clouds (mainly, cirri and cirrostratus) sometimes show rings and arcs of light (often colored) around the Sun or the Moon, as shown in Fig. 1. These optical phenomena are called halos.[1]Halos can be of many types, with each type corresponding to the shape and orientation of ice crystals in clouds (Greenler, 1980). Halos provide valuable information about the atmosphere. Observing halos from the ground is an important way to understand meteorological processes. It is a wide-area observation of ice crystals in clouds. For example, the difference in frequency of appearance between 22° and 46° halos suggests the ratio of pristine to non-pristine crystals in clouds (van Diedenhoven, 2014). There are also several studies of ice crystals and halo observations (Lynch and Schwartz, 1985; Sassen et al., 1994; Um and McFarquhar, 2015; Sassen, 1980; Lawson et al., 2006). An attempt is also being made to achieve automated observations of halos using a sun-tracking camera (Forster et al., 2017), or a Total Sky Imager (Boyd et al., 2019). Image processing techniques can be used to observe even faint halos or other atmospheric optical phenomena in photographs more clearly, which can greatly aid in these types of studies (Riikonen et al., 2000; Moilanen and Gritsevich, 2022; Großmann et al., 2011).

A halo is formed by the interaction of sunlight with the particles in a cloud. However, the contrast of the cloud itself prevents us from seeing halos clearly. To resolve this difficulty, halo observers have developed various image processing techniques that enhance halos in images and improve their observation, and a B−R (blue minus red) processing proposed by Rossetto (Rossetto, 2011; Lefaudeux, 2013) is one such technique and is known to be an effective method for enhancing halos. It involves subtracting a pixel value in the red channel from that in the blue channel and setting the difference as the gray value of the pixel. As a result, the areas in an image that are more reddish or bluish than the surrounding areas are enhanced. The author

---

[1]'Halos' are typically used to refer to sun-centered rings, while 'arcs' refer to the other type of atmospheric phenomena caused by ice crystals. However, in this manuscript, we will use 'halos' to refer to both.

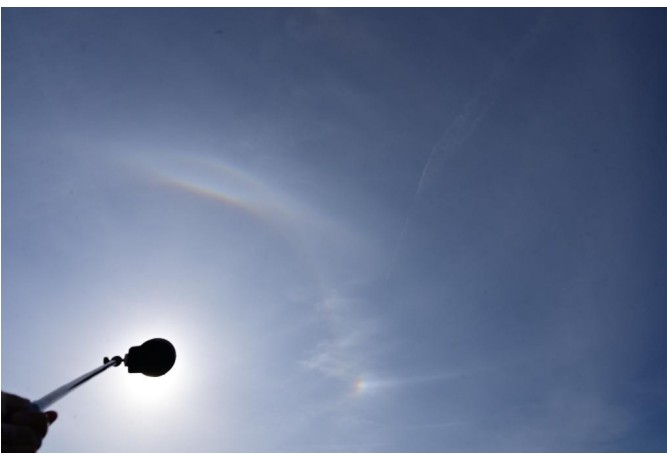

**Figure 1.** Types of halos: This image contains at least nine types of halos. (All the photographs in this manuscript were taken by the author in Tokyo, Japan.)

of the present study has developed a revised method, called autoBR, which was implemented in an image processing tool, named Atmospheric Optical Image Enhancer (Ayatsuka, 2022). In autoBR, the red and blue channels are differently weighted and the green channel is also referenced (details are described in Section 3). In contrast, in B−R processing, the red and blue channels are equally weighted and the information in the green channel is not used.

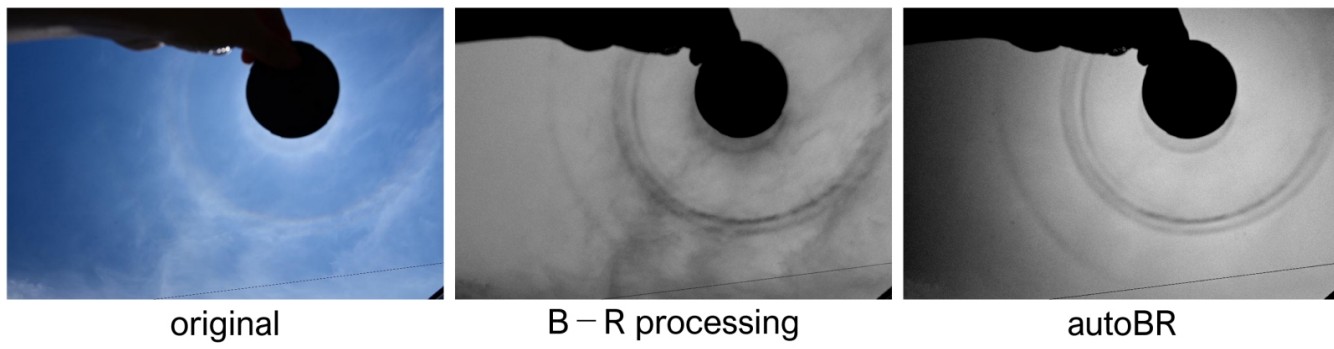

original     B−R processing     autoBR

**Figure 2.** Image results obtained using of the B−R processing and autoBR

Fig. 2 shows a halo image processed with both B−R processing and autoBR. It is quite noticeable that the contrasts of
30 the clouds are flattened by the two processes, especially by autoBR. In other words, these processes are thought to flatten the "cloud colors" in the image, and extract the appearance and intensity of halos. It is quite useful for observing and analyzing halos precisely through ground images.

This study constructs and validates a sky-color model for halo display. Using the model, a new algorithm is developed, called sky-color regression, to more effectively cancel cloud colors and enhance halos in the image.

The rest of the paper is organized as follows. Section 2 discusses related work on image processing for similar purposes. Section 3 presents details of existing algorithms, B−R processing, and autoBR. Section 4 proposes a sky-color model and validates it with an image of a halo display. Section 5 describes the new algorithm called sky-color regression. Section 6 discusses some remaining issues related to the new algorithm and Section 7 concludes the paper.

## 2    Related Work

As described in the previous section, the contrasts of clouds should be reduced to extract halos in an image. Similar processes have been studied for other purposes, for example, for dehaze or fog reduction. Cloud detection using color information obtained from images is another related research topic, although with an opposite pupose.

### 2.1    Dehaze/Fog Reduction

The main goal of dehaze/fog reduction is to compensate the poor camera-image quality when observing land and sea scenes
under hazy conditions (Chengtao et al., 2015; Singh et al., 2016). Building physical models of the hazed view to recover background information is one of the approaches that is similar to the one used in the present study. Shi et al. (2022) used blue and red channel information to improve images in sandstorms.

The main difference between these studies and the present one is that, in their target situation of the previous studies, complex background images are obscured by smooth haze or fog, whereas in the target situation of the present study, relatively simple
patterns are obscured by clouds. Because the present study has a different purpose, which is to process the image for easier halo detection and not to compensate for the image quality, a simpler model than those described in previous studies can be employed.

Gao and Li (2017) described the removal of cirrus clouds from the ground images taken by a Landsat satellite by using a microwave imager. An image taken by a band that is highly reflected by cirrus clouds helps in effectively removing cirrus
clouds from the corresponding visible light (RGB) image.

### 2.2    Cloud Detection

Many studies have investigated the detection of cloud regions, or segmentation in sky images taken from the ground for meteorological observation. Koehler et al. (1991) set a threshold on the ratio of red and blue pixels that can be utilized to judge whether a pixel is a part of clouds.
Dev et al. (2017) investigated the contribution of composite parameters for a cloud segmentation algorithm, including values in color spaces other than RGB, such as HSV and L*a*b. They found that the saturation in HSV, the ratio of red and blue in RGB, and $\frac{B-R}{B+R}$, also called the "normalized blue/red ratio" in Li et al. (2011), are important for cloud detection. These studies suggest that the blue/red ratio is also important for cloud cancellation and halo extraction in an image.

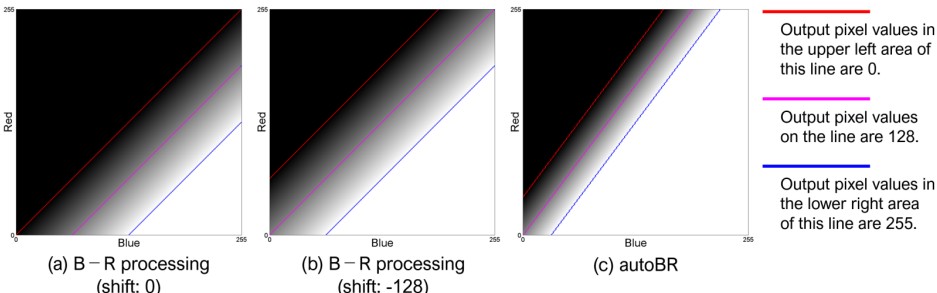

**Figure 3.** Color-to-gray translation maps for B−R processing and autoBR, in the B vs R space: the magenta lines show the center value (128), and the blue and the red lines show the saturation points (255 and 0).

## 3 Existing Algorithms

This section describes the existing algorithms for extracting halos in images, i.e., B−R processing and autoBR. It also shows how color pixels are translated into grayscale pixels in translation maps.

### 3.1 B−R processing

B−R processing, also referred as "color subtraction," is a widely used technique for enhancing and explaining halos and other atmospheric optical phenomena[2]. It is introduced as an image processing technique using a tool called the "channel mixer." It is

70 available in PhotoShop or such other photo- retouching applications. The channel mixer outputs a new pixel value of a specified channel calculated from the RGB values of the corresponding pixel in a source image. The values are added or subtracted with specified weights, and a "shift" value is used to adjusts the output range to keep the pixel values in the interested area of interest within 0-255.

For B−R processing, the weight for a red channel is set to the same magnitude but with opposite sign as the weight for a blue

channel. Although the magnitude is adjustable, it is often set to 200%, the maximum value allowed by the tools. The shift value is set and adjusted by the user by looking at the resulting image. Fig. 3(a) shows a resulting gray map when the parameters are set to B: 200%, G: 0%, R: −200%, and shift: 0. The blue/red lines show the edge of the underexposed/overexposed area, respectively, while the magenta line shows pixels in the center value (128). The map depicted in Fig. 3(b) shows the case when the shift parameter is set to −128.

The pixel-value translation in B−R processing is represented as:

$$I = \alpha(B - R) + \beta \tag{1}$$

where $I$ is the output intensity value, $B$ and $R$ are pixel values in the source image, and $\alpha$ and $\beta$ are the "magnitude" and "shift" ($I$ is set to 0 if $I < 0$, and to 255 if $I > 255$). For larger $\alpha$, the contrast of the output image will be higher and the width of the area between the blue and red lines in a gray map will be narrower.

---

[2]See https://atoptics.co.uk/blog/opod-helic-lowitz-arcs-france/ or https://atoptics.co.uk/blog/opod-helic-lowitz-arcs-france/ for examples.

 ## 3.2  autoBR

Extending B−R processing, more effective parameters have been heuristically explored and the automatic adjustment of the "shift" parameter has been implemented for the autoBR algorithm. The pixel-value translation in autoBR is represented as:

$$I = \alpha(\omega_B B + \omega_G G - R) + \beta \tag{2}$$

where $\alpha = 2.0$, $\omega_B = 1.5$ and $\omega_G = 0.25$. The $\beta$ is set to a value such that the average of the $I$ values (excluding too dark-/bright areas in the source image) becomes 128. An example gray map for the translation is shown in Fig. 3(c), where the blue/red/magenta lines are steeper than those for B−R processing.

## 4   Model and Validation

Considering how the presented algorithms work effectively to extract halos, a model of sky-color in images can be assumed. This section explains the model and validates it with some photographs.

### 4.1   A Model for Sky Color

This study provides a simple physical model for celestial light, with no direct beam from the light source and no particles other than those that form clouds. Light $L_{sky}$ from one direction in the sky contains following (Fig. 4):

1. Rayleigh scattering $L_R$ from air molecules (sky blue background)

2. Mie scattering $L_M$ from cloud particles (white cloud), and

3. Refracted/reflected light $L_H$ from cloud particles (halos, etc.)

In a digital image, $L_{sky}$, $L_R$, $L_M$, and $L_H$ are vectors typically containing $R$, $G$, and $B$ intensities. If translation $g$ that converts colors in an image to grayscales reduces the contrast of the cloud color, it satisfies the following formula:

$$g(L_R + L_M + L_H) \approx g(L_R + L_H) \tag{3}$$

Considering the color-to-gray translation maps in Fig. 3, the above formula implies that the direction of vector $L_M$ is almost parallel to the red/blue/magenta lines when projected onto the B vs. R space. Fig. 5 shows the model for sky color in the B vs. R space. $L_R$ in the Fig. 5 represents sky color if the clouds and halos are not there. The arrows represent vectors, $L_M$ and $L_H$, added to the vector $L_R$ (in the B vs. R. space). The color of the sky at a given point in an image is the result of the summation of three vectors.

### 4.2   Validation of the Model

Fig. 6 shows a halo display image and its color maps of it in B vs. R, R vs. G, and G vs. B spaces. The pixel colors on the maps other than black represent (one of) the real colors in the image. The colors are distributed linearly on the maps, roughly parallel

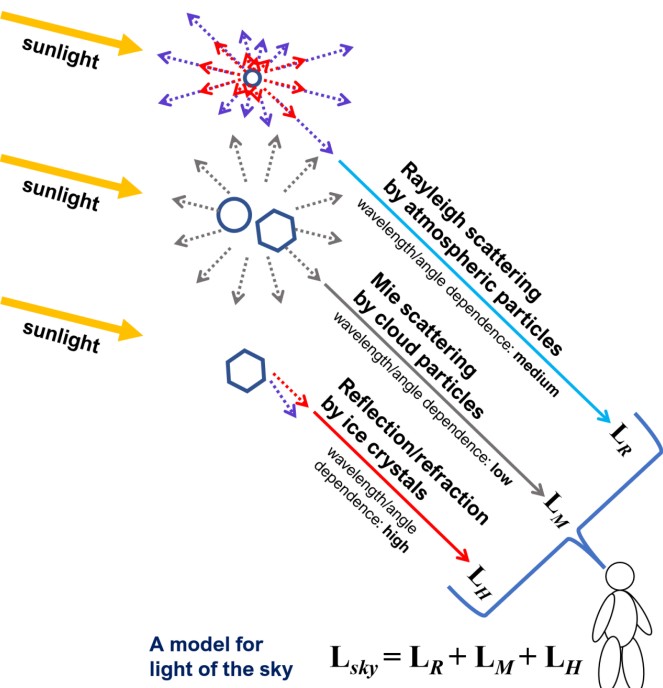

**Figure 4.** A physical model of the components of skylight

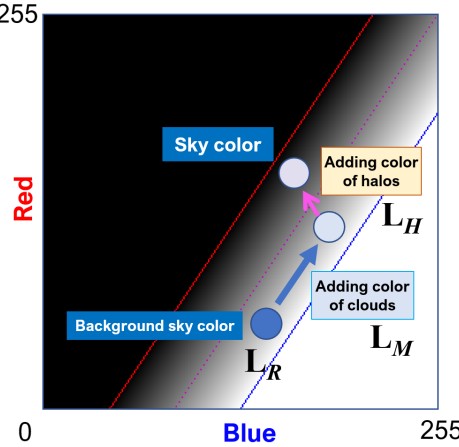

**Figure 5.** A pixel color model of skylight on an image: The arrows represent vectors, $L_M$ and $L_H$, added to the vector $L_R$ (in the B vs. R. space). The color of the sky at a given point in an image is the result of the summation of three vectors.

to the lines shown in Fig. 3. Some colors are outside the main distribution, corresponding to the silhouette of the foreground objects in the image. The width of the main distribution is as narrow as the area between the blue and red lines depicted in Fig. 3. These reasons explain the effectiveness of B−R processing and autoBR.

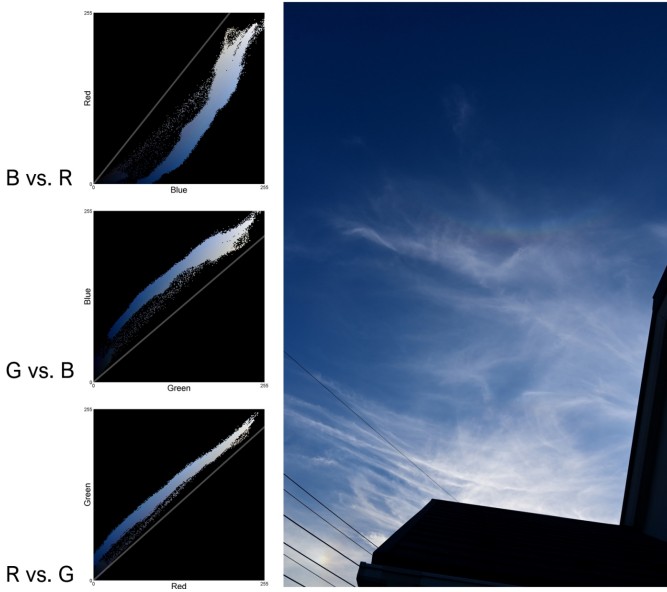

**Figure 6.** Color maps of a sky image including halos

The lines in the color maps are regression lines for the distributions of color in each space. From the slope of the B vs. R regression line, it can be expected that autoBR with a higher slope would be more effective than the B−R method with a slope of 1.0. Blue minus green is a variation of B−R processing(Juutilainen, 2015) and will be effective when the slope of the regression line in the B vs. G space is close to 1.0.

Figs. 7 and 8 show partial color maps and their regression lines for the areas indicated by box lines. The partial color maps, corresponding to the red and yellow boxes, are superimposed on the full color maps shown in dimmed colors.

Fig. 7 shows the color maps and regression lines for areas without halos. These maps are thinner and more linear than the full map. This means that the blue of the sky, whiteness of the light, and shadow of the clouds are also uniform in these small areas. The slopes of the regression lines for the red and yellow boxes are clearly different from each other, especially in the B vs. R space. As a result, the optimal parameters for reducing cloud contrast vary from region to region.

Fig. 8 shows the color maps of the areas with halos. The red box contains a left parhelion and the yellow box contains part of a circumzenithal arc. Notably, the loop part of the color map seems to correspond to the parhelion. Also note that there is a bump in the lower part of the B vs. R color map, indicating the red color of the circumzenithal arc. Such features are less significant in the G vs. B space and somewhat difficult to distinguish in the R vs. G space.

Color maps for other halo display images have similar characteristics, which supports the validity of the sky color model shown in Fig. 5.

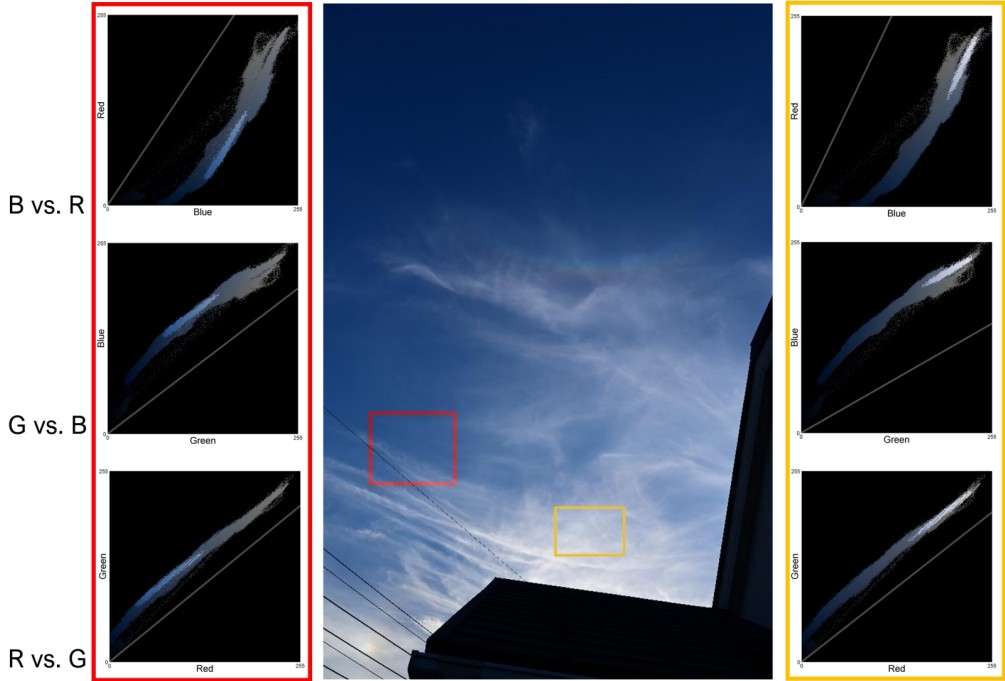

**Figure 7.** Color maps of areas without halos in the sky image: Color maps for the red box are shown in the left column, and those for the yellow box are shown in the right column.

## 5  Sky-Color Regression Algorithm

The color maps explained in the previous section suggest that an algorithm can be revised for reducing cloud contrast by adjusting the parameters according to the slope of the regression line. The maps also suggest that the parameters should be adjusted locally, rather than using one set of parameters for all areas of an image. Based on these observations, a new algorithm called 'sky-color regression' is constructed.

### 5.1  Regression-based Processing

Let $\omega_{BR}$, $\omega_{RG}$, and $\omega_{GB}$ denote the slopes of the regression lines in B vs. R, R vs. G, and G vs. B spaces. Herein, a transformation from the RGB color to grayscale is considered as follows:

$$I = \alpha_1(\omega_{BR}B - R) + \alpha_2\omega_{RG}\omega_{GB}G + \beta \tag{4}$$

For this study, the following conditions were set for the new algorithm: $\alpha_1 = 2.5$ and $\alpha_2 = 1.0$. However, it has not yet been discussed how the green channel affects the sky color; therefore, the values were explored heuristically. A value of $\beta$ was set as in autoBR, i.e., the average of $I$ should be 128. This process is called "regression-based processing."

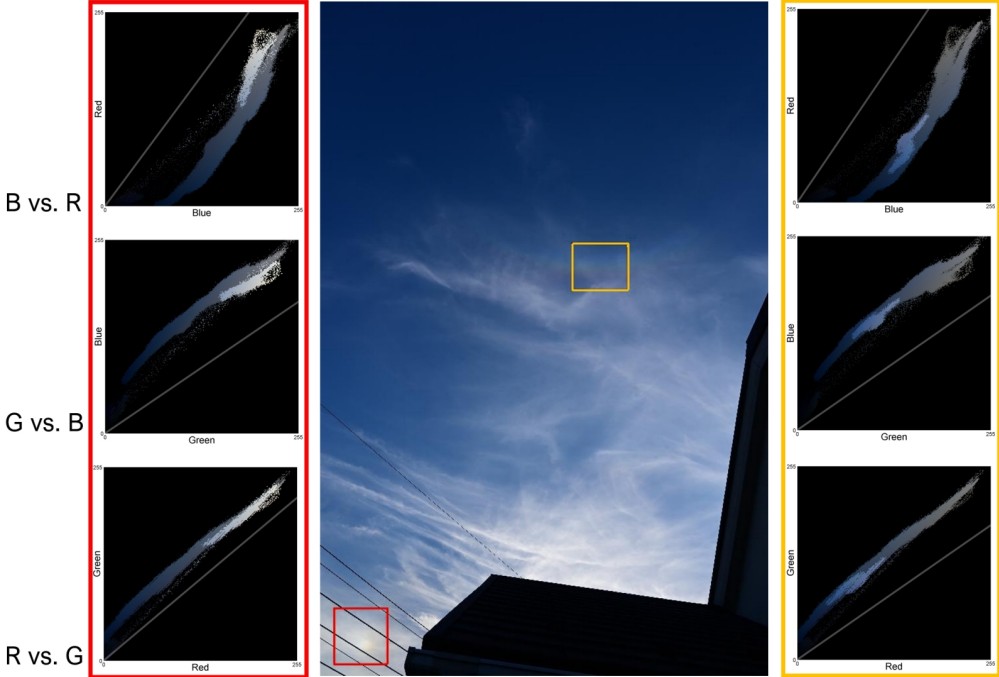

**Figure 8.** Color maps of areas with halos of the sky image: Color maps for the red box, including a parhelion, are shown in the left column, and the those for the yellow box, including a circumzenithal arc, are shown in the right column.

Fig. 9 shows the images processed with autoBR and regression-based processing. When regression-based processing is applied to the entire image, clouds around the circumzenithal arc are reduced more than that achieved with autoBR, although the former method seemed to be less effective for the other areas. When regression-based processing was applied to a part of the image, clouds in the part were effectively reduced, and the brightness is also adjusted accordingly.

## 5.2 Local Adaptive Processing

If the area to calculate the regression line is too small, the accuracy of the line will worsen. However, if the area is too large, the locality will deteriorate. Assuming a wide-angle image such as the photos shown in this paper, the default value was set to $1/6^{th}$ of the long side of an image.

For best results, regression parameters $\omega_{BR}$, $\omega_{RG}$, and $\omega_{GB}$ should be updated for every pixel in an image; however, this is computationally expensive. If the best parameters for neighboring pixels are similar, some calculations can be omitted as shown below.

1. The parameters are updated per block of a few pixels.

2. The regressions are performed on an appropriately shrunken image.

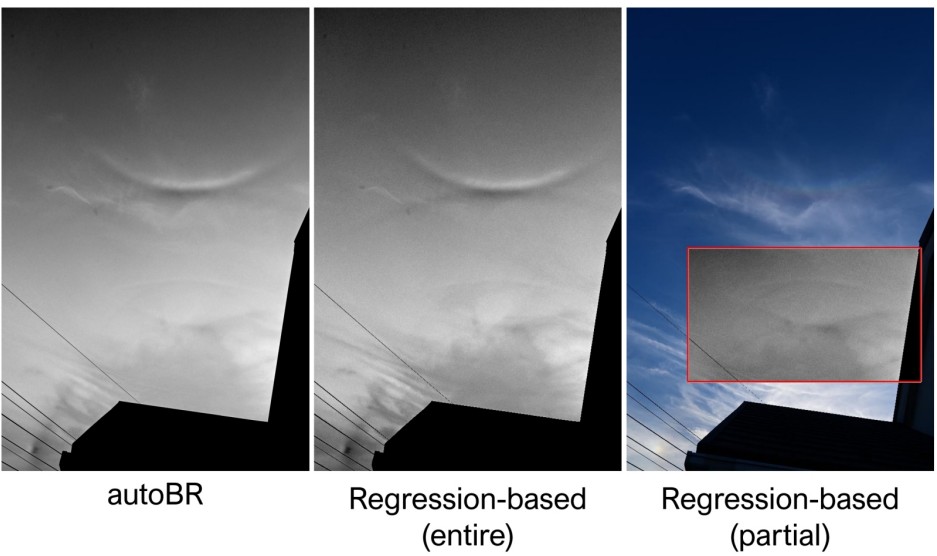

| autoBR | Regression-based (entire) | Regression-based (partial) |

**Figure 9.** Comparison of regression-based processing with autoBR

While implementing of the sky-color regression algorithm, a target image should be divided into $b \times b$-pixel blocks. The regression parameters are calculated on a shrunken image of size $1/b$. The default value of $b$ is set so that the long side of the shrunken image is approximately 1000 pixels, i.e., $b = \lfloor p/1000 \rfloor$, where $p$ is the long side of the target image. If the camera angle is known, the value can be optimized using this information. How to calculate the best value for each angle is a future work of this study.

An example of an image processed by the sky-color regression algorithm is shown in Fig. 10. The width and height of the image were $4024 \times 6048$ pixels, and the image was processed in 4.3 s using sky-color regression implemented in Java on a PC equipped with Intel Core i7-1165G7 (2.80 GHz).

More examples are shown in Fig. 11, which compares sky-color regression with B−R processing and autoBR. While the algorithms mainly enhance colored halos, a colorless parhelic circle is also visible in the image at the second row from the bottom. It suggests that a parhelic circle appears slightly bluer than the background clouds.

## 6    Discussion

### 6.1    Evaluation

Fig. 12 displays histograms of processed image parts by the sky-color regression, B−R processing and autoBR. Rows (a) and (c) show areas without halos, while row (b) shows an area with circumzenithal and supralateral arcs.

With the sky-color regression and autoBR, distributions of pixel values for areas (a) and (c), without halos, are simple standart distributions, while there are two or more peaks in distributions with B−R processing. Standard deviations are also

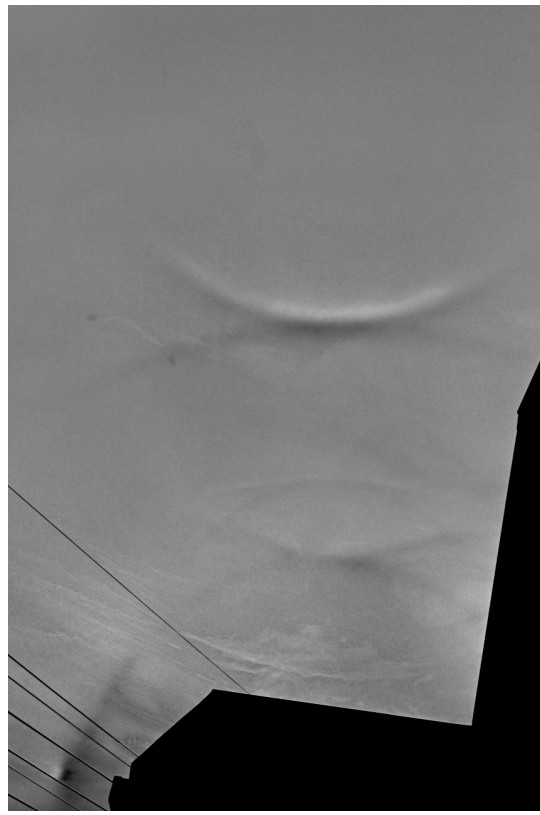

**Figure 10.** Processed with sky-color regression: In the original image (see Fig. 6), even experienced observers can hardly detect halos except for a parhelion, a circumzenithal arc, and a part of the 22° halo and the upper tangent arc.

smaller with the sky-color regression than with B−R processing. For area (a), the standard deviation is smaller when using the sky-color regression compared to autoBR. Conversely, for area (c), the standard deviation is smaller with autoBR than with the sky-color regression. It shows that the sky-color regression is not always the most effective method for canceling out cloud, and to be refined in future studies.

For area (b), all algorithms produced a histogram with three peaks corresponding to the clouds, reddish parts, and bluish parts of the halos. However, the peaks are most clearly separated with the sky-color regression. The standard deviation is also the largest with the sky-color regression.

The average values for the areas are maintained around 128, which is the midpoint value of the range 0 to 255, with the sky-color regression. It shows that the local adaptive processing works.

## 6.2   Other Atmospheric Optical Phenomena

The sky-color regression algorithm can also be used efficiently to enhance other colored atmospheric optical phenomena. Fig. 13(a) shows an example of quite faint third and fourth order rainbows. The algorithm extracts colored bows from the

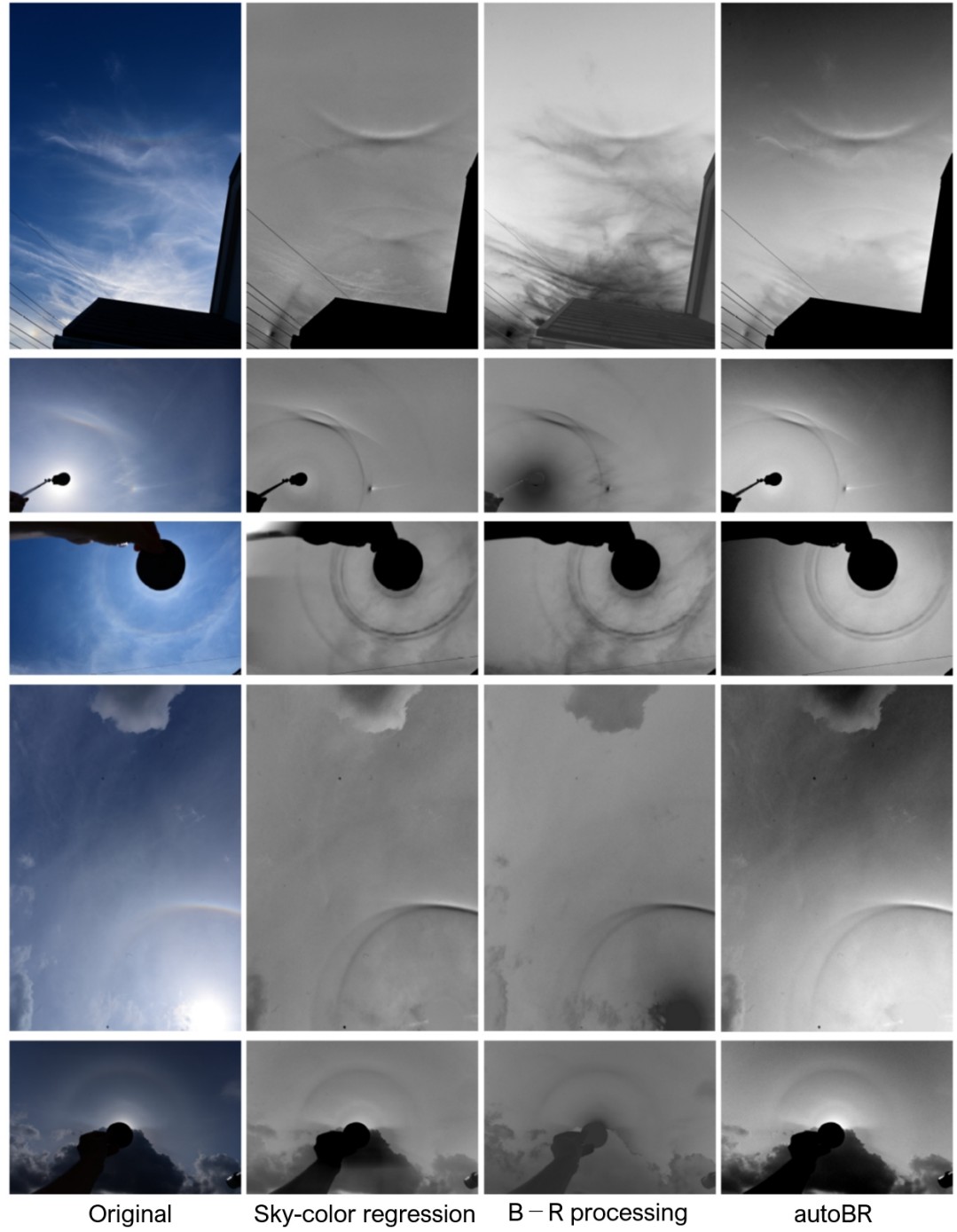

| Original | Sky-color regression | B − R processing | autoBR |

**Figure 11.** Examples of sky-color regression in comparison to B−R processing and autoBR

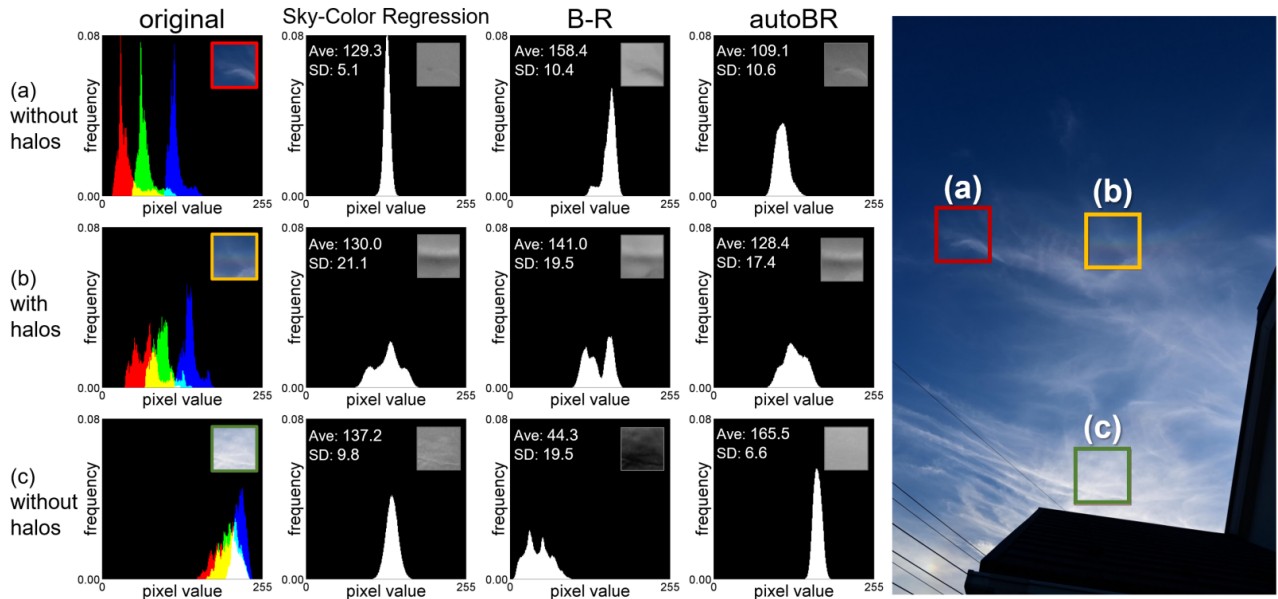

**Figure 12.** Histograms of processed image parts: (a) and (c) are areas without halos, while (b) is an area with circumzenithal and supralateral arcs.

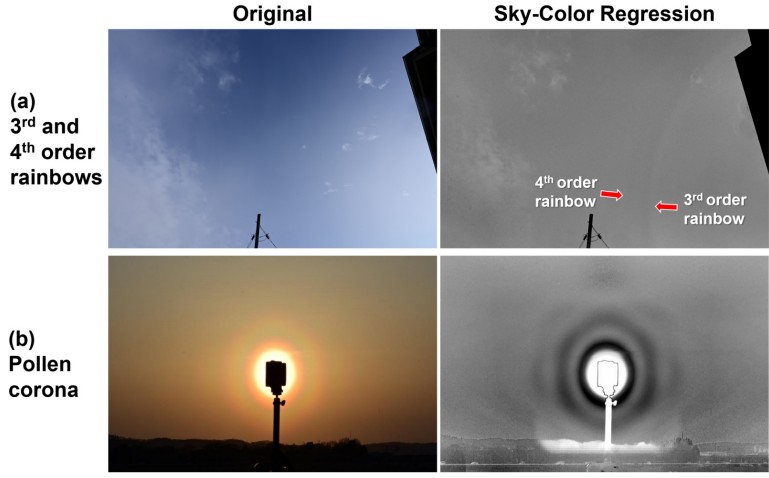

**Figure 13.** Applying sky-color regression for images of other atmospheric optical phenomena: (a) third and fourth order rainbows, (b) pollen corona (Japanese ceder).

background sky without clouds. Fig. 13(b) shows the diffraction pattern of Japanese cedar pollen corona. The algorithm also works on the sky color near sunset.

## 6.3 Limitations

In the sky-color regression algorithm, the mixing of the red and blue channels is calculated statistically, while the weight of the green channel is adjusted heuristically. The size of the pixel block and the area over which the regression is performed are also determined by heuristics. These parameters would also be statistically determined if further analysis was performed on the images of halo displays.

The color maps of an image change according to the white balance setting. Because of its adaptive nature, sky-color regression is less sensitive to white balance than $B-R$ processing and autoBR are. Because digital cameras often have more color settings than white balance, how the settings affect the results of the algorithms need to be examined.

The sky-color regression algorithm has been developed and optimized mainly for the images captured by standard digital single-lens reflex (DSLR) cameras equipping lenses with focal length 24 mm to 100 mm. Adjusting parameters may be necessary for other types of images, such as total sky images. Optimizing and evaluating the algorithm for total sky images is an area for further study.

## 7 Conclusions

This paper considered an image processing method to extract atmospheric optical phenomena such as halos by reducing the contrast of clouds according to their color in the sky image and constructed a model of sky color. It also validated the model with halo images and implemented a new algorithm named sky-color regression. The algorithm extracts halos, including those that are difficult to distinguish with the naked eye. Even the most popular 22° halos are sometimes camouflaged by clouds. In such cases, the new algorithm might allow automated observations. In processed images, halos, which are often geometrically shaped, are more easily detected by feature detection algorithms. The algorithm is useful not only for accurately observing the appearance of halos, but also to measure the intensity of halos by removing cloud contrasts.

Although the purpose of this work was to process sky images for easier halo detection, the model constructed and validated in this paper and the method developed can be used for other purposes as well such as cloud detection. Similar models and processing can be devlleoped that would benefit other fields.

*Code availability.* Sample codes for Sky Color Regression in Java are available on https://doi.org/10.5281/zenodo.7716821

*Author contributions.* All of the contents in this manuscript including photographs, analyses, algorithms, sample codes, were prepared by YA.

*Competing interests.* The author declare that they have no conflict of interest.

*Acknowledgements.* I thank Dr. RainbowMustache (@HIGEHIGErainbow on Twitter) to introduce me B−R processing, and Dr. Kentaro Araki to advise me where to submit this paper.

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
