# Peer review of "Revealing Halos Concealed by Cirrus Clouds"

_EGUsphere, 2023_

## Author Comment (AC1)

**Response to RC2 (Anonymous Referee #4)**

**Revealing Halos Concealed by Cirrus Clouds**

Yuji Ayatsuka

I would like to thank the reviewers for their kind and useful comments and constructive suggestions on our manuscript. We have modified the manuscript according to the comments as possible. The modifications in our revised manuscript are listed below.
* * *
**Main comment**

- *The major revision I would like to see is a quantitative assessment of the Boyd (or other) method using the four image processing techniques described here. To save time, the Boyd training set (presumably available online or from the authors) could be used as a benchmark. Can you show quantitatively that there is an advantage to the new approach?*

Thanks for the insightful comment. Boyd and Foster's algorithms are designed to detect halos in an image, while ours is designed to process images into ones in which we can easily detect halos, making it difficult to compare them directly at this time. Although our algorithm is able to make Boyd and Foster's algorithms more efficient, I think this will be one of our future studies. Instead of the comparision to their algorithms, as a quantitative evaluation of the algorithms presented in the manuscript, I have added Fig. 12, which shows histograms of the processed images and some explanations of them.

Applying our algorithm to Total Sky Image datasets will also be an interesting future study, but we may need some more parameter optimization of the algorithm, since we mainly tuned parameters for images taken with standard cameras and lenses. I have also added some descriptions about this.

[Figure]

**Figure 12:** Histograms of Processed Image Parts: (a) and (c) are areas without halos, while (b) is an area with circumzenithal and supralateral arcs.

**Line 152: 6.1 Evaluation**

Fig. 12 displays histograms of processed image parts by the sky-color regression, B−R processing and autoBR. Rows (a) and (c) show areas without halos, while row (b) shows an area with circumzenithal and supralateral arcs.

With the sky-color regression and autoBR, distributions of pixel values for areas (a) and (c), without halos, are simple standart distributions, while there are two or more peaks in distributions with B−R processing. Standard deviations are also smaller with the sky-color regression than with B−R processing. For area (a), the standard deviation is smaller when using the sky-color regression compared to autoBR. Conversely, for area (c), the standard deviation is smaller with autoBR than with the sky-color regression. It shows that the sky-color regression is not always the most effective method for canceling out cloud, and to be refined in future studies.

For area (b), all algorithms produced a histogram with three peaks corresponding to the clouds, reddish parts, and bluish parts of the halos. However, the peaks are most clearly separated with the sky-color regression. The standard deviation is also the largest with the sky-color regression.

The average values for the areas are maintained around 128, which is the midpoint value of the range 0 to 255, with the sky-color regression. It shows that the local adaptive processing works.

**next to Line 158:** The sky-color regression algorithm has been developed and optimized mainly for the images captured by standard digital single-lens reflex (DSLR) cameras equipping lenses with focal length 24 mm to 100 mm. Adjusting parameters may be necessary for other types of images, such as total sky images. Optimizing and evaluating the algorithm for total sky images is an area for further study.
* * *
- *More justification of the relevance of detecting halos is needed. Many studies, perhaps some by Alexei Korolev (ECCC), Greg McFarquhar (Univ. Oklahoma), Ben Murray (Leeds), Sergey Matrosov (Univ. Colorado), Knut Stamnes (SIT), Ping Yang (Texas A&M), and others could be used to motivate the importance of information about ...*

Thanks for listing related researches. I have added some references and description in the introduction.

**Line 10:** It is in an in situ and wide-area observation of ice crystals in clouds. For example, the difference in frequncy of appearance between 22° and 46° halos suggests the ratio of pristine to non-prestine crystals in clouds (van Diedenhoven, 2014). There are also several studies ice crystals and halo observations (Lynch and Schwartz, 1985; Sassen et al., 1994; Um and McFarquhar, 2015; Sassen, 1980; Lawson et al., 2006).
* * *
- *More review of how halo images are analyzed would benefit the paper.*

I have added references of the papers conatining images processed by some techiniques. (Updated references are listed at the end of this letter.)

*Line 12:* Image processing techniques can be used to observe even faint halos in photographs more clearly, which can greatly aid in these types of studies (Riikonen et al., 2000; Moilanen and Gritsevich, 2022; Großmann et al., 2011).

- *More review of how the older B-R technique is used would also benefit the paper.*

I would like to refer to papers that describe and analyze the B−R processing in detail, but I could not find them (it is one of the motivations for writing this manuscript). I have added quantitative comparizons of the B-R and other techiniques as Figure 12, and also added footnotes at the description of the B-R processing.

**Line 59:** B−R processing also referred as "color subtraction," is a widely used technique for enhancing and explaining halos and other atmospheric optical phenomena.
*Footnote for it:* See
https://atoptics.co.uk/blog/opod-helic-lowitz-arcs-france/ or
https://atoptics.co.uk/blog/opod-helic-lowitz-arcs-france/ for examples.

**Minor comments**

- *Line 8. Somewhat nitpicky, but halos refer to full circle displays (implying randomly oriented particles) and oriented displays are arcs; i.e., Fig. 1 includes both arcs and halos.*

Sorry for the ambiguous words, but a term "halos" sometimes includes arcs in papers on atmospheric optical phenomena such as "Atmospheric Halos" by Walter Tape. I have added description and a footnote that clearly define "halos" including arcs in this manuscript, in the abstract and in the main texts.

**Line 1:** ... halos (including arcs) appear in the sky.
*Footnote for Line 8:* 'Halos' are typically used to refer to sun-centered rings, while 'arcs' refer to the other type of atmospheric phenomena caused by ice crystals. However, in this manuscript, we will use 'halos' to refer to both.

- *Line 42, 165. The purpose isn't to detect halos, right? Rather, the purpose is to process images so as to enhance light associated with halos and increase the accuracy of detection algorithms. See Major comment (a).*

Thanks for the comment and sorry for the ambiguous notation. Of course, the purpose of the algorithms is to process image for easier halo detection. I have modified some descriptions.

**Line 42:** "to detect halos" → "to process the image for easier halo detection"
**Line 165:** "to extract halos from sky images" → "to process sky images for easier halo detection"

- *Lines 18-20: I can't tell if autoBR needs a citation or if it is being introduced in the present manuscript.*
- *Line 76: Explored heuristically where? Earlier you stated that you developed it but no reference was provided. Is this being introduced here or can it be referenced?*

Sorry for unclear descriptions. I implemented the autoBR algorithm in a tool (called Atmospheric Optical Image Enhancer (AOI, for short)) that was released two years ago, but details of the algorithm are explained first in this manuscript.

- *Line 26: It seems like this technique would work best for arcs that are refractive (i.e., disperse white light into spectral constituents like a 22 deg halo) but not for arcs that are reflective (i.e., are also white, like a parhelic circle). Is that true?*

It is almost true. However, the algorithms also work on some colorless arcs, such as a parhelic circle (see the images in the second row from the bottom in Figure 11). In another example not in this manuscript, a helic arc can be seen in the processed image. From these results we can know that such arcs are somehow bluer than the background clouds. It is an interesting point that the algorithms allow such an analysis.

I have added a description about it in the manuscript.

**Line 149:** While the algorithms mainly enhance colored halos, a colorless parhelic circle is also visible in the image at the second row from the bottom. It suggests that a parhelic circle appears slightly bluer than the background clouds.
* * *
- *Figure 5, Line 95: I'm having a trouble understanding what is being represented here. The annotation is ambiguous and the arrows aren't explained. Can you add information to the caption and add interpretation to the text?*

Sorry for the unclear explanations. The "colors" on the picture are some kind of vectors consisting of R, G and B values. The arrows represent the addition of vectors. I have added the explanations in the caption of Figure 5.

**Figure 5:** The arrows represent vectors, $L_M$ and $L_H$, added to the vector $L_R$ (in the B vs. R. space). The color of the sky at a given point in an image is the result of the summation of three vectors.
* * *
- *There are no photo credits. Were the photos all taken by the author? If they aren't, please add credits. If they are, congrats on the odd radius halos in the middle row of Fig 11. Outstanding capture!*

Yes, all the photographs in this manuscript were taken by the author. I have added the description in the caption of Figure 1. (Thanks for the comment,

and I am glad you like the photo of the pyramidal halo display!)

**Figure 1:** (All the photographs in this manuscript were taken by the author in Tokyo, Japan.)
* * *
- *Typos*

Thanks for catching these typos. I have corrected them all.
* * *
**References**

[revised manuscript text omitted]

---

## Author Comment (AC2)

**Response to RC1 (Annonymous Referee #3)**

**Revealing Halos Concealed by Cirrus Clouds**

Yuji Ayatsuka

I would like to thank the reviewers for their kind and useful comments and constructive suggestions on our manuscript. We have modified the manuscript according to the comments as possible. The modifications in our revised manuscript are listed below.
* * *
**Main comment**

- *However, it remains unclear why this is even important, and there should be more explanation why this work is important in the introduction. It would also be valuable to discuss if the algorithm is then used to determine the kind of halo as well as ice crystal habits typically associated with halo types etc. As it is, the text leaves the reader wonder why even care about it, especially since the halo is dependent on sun and camera positioning.*

Thank you for the comment. I have added some descritions and references to explain the importance of halo observations on photographs in the introduction and the conclusions. (Updated references are listed at the end of this letter.)

**Line 11:** It is in an situ and wide-area observation of ice crystals in clouds. For example, the difference in frequncy of appearance between 22° and 46° halos suggests the ratio of pristine to non-prestine crystals in clouds (van Diedenhoven, 2014). There are also several studies ice crystals and halo observations (Lynch and Schwartz, 1985; Sassen et al., 1994; Um and McFarquhar, 2015; Sassen, 1980; 15 Lawson et al., 2006).

**Line 31:** ... and extract the appearance and intensity of halos. It is quite useful for observing and analyzing halos precisely through ground images.

*Line 199:* In processed images, halos, which are often geometrically shaped, are more easily detected by feature detection algorithms. The algorithm is useful not only for accurately observing the appearance of halos, but also to measure the intensity of halos by removing cloud contrasts.

**Minor comments**

- *Line 19: AOI is not explained*

I'm sorry for the malformatted citation. I have revised the description and the citation.

**Line 18:** The author of the present study has developed a revised method, called autoBR, which was implemented in an image processing tool, named Atmoephric Optical Image Enhancer (Ayatsuka, 2022).

- *Line 19: "differently weighted" - how exactly?*

It is described in the later section in the manuscript, so that I have added some words to mention it.

**Line 19:** In autoBR, the red and blue channels are differently weighted and the green channel is also referenced (details are described in Section 3).

- *Equation (3): What exactly does the equal symbol with dots stand for? Many readers will not be familiar with that symbol since it' s not commonly used (I have not seen it before).*

I used a symbol "equal with dots" ($\fallingdotseq$) for meaning "approximately equals". I haven't known, but it is an East Asian local style. Thank you for pointing it out. I have replaced it with '$\approx$'.

**Equation (3):** $g(L_R + L_M + L_H) \approx g(L_R + L_H)$

- *Line 145: Why is "b" chosen depending on the pixel number of an image? Why not depending on the camera angle? If the camera angle is wider, there would be more variations in color across the image.*

Thanks for the important comment. I set the current value of parameter "b" only as a default setting to balance quality and processing time, and for the case when camera angle information is missing. As reviewer #3 mentioned, optimizing the parameter "b" according to the camera angle will bring better quality. I think how to optimize it is one of the future studies of this work. I have added two sentences to describe it.

[revised manuscript text omitted]

---

## Author Response (AR2)

**Response to Reviewers (2)**

**Revealing Halos Concealed by Cirrus Clouds**

Yuji Ayatsuka

I would like to thank again the reviewers for their kind and useful comments and constructive suggestions on our manuscript. The modifications in our revised manuscript are listed below.

**To Anonymous Referee #3**

- *The manuscript "Revealing Halos Concealed by Cirrus Clouds" by Y. Ayatsuka presents a new algorithm to automatically detect halos in images that are often hard to see. Generally, the manuscript is well written and fits well within the scope of AMT. The author has addressed my comments satisfactorily and I suggest publication after editing of some typos found in the document.*

Thanks for your comment. I have fixed all the typos you found.

**To Anonymous Referee #4**

- *I understand that the Boyd and Foster algorithms are detection methods whereas the method in the paper currently under consideration is designed to improve the image processing. My initial criticism was that the proposed processing techniques are evaluated qualitatively using a small number of hand-picked example images and that even amongst those images, the advantages are inconsistent (e.g., Fig. 11). Boyd et al. reports that their detection method is 86% accurate relative to a curated training set. My suggestion was to run the Boyd et al. algorithm after first processing the images using the methodology presented here to test the hypothesis that the new processing approach will increase the accuracy of detection.*

Thanks for the comment. Running the Boyd et al. algorithm on images pre-processed by our algorithm is quite interesting, but I think it is beyond the scope of this manuscript for several reasons. First, Boyd's algorithm requires images in which each pixel represents an intensity, while our algorithm outputs images in which each pixel represents a kind of color difference signal. Then we have to modify and tune Boyd's algorithm to apply it to the

output images. Second, as I mentioned in the first response, our algorithm is currently tuned for images taken with standard cameras and lenses. I also need to adopt some parameters of our algorithm for total sky images to apply Boyd's algorithm.

As shown in Fig.12, the pixels of redder and bluer part of halos are clearly separated (color differences are large) by our algorithms, while the pixels of clouds are concentrated in narrow range of values (smaller color differences). It obviously helps the development and revision of halo detection algorithms, including Boyd's one, much easier. It is also quite helpful for human observers to find and distinguish many types of halos in the image, some of which are often difficult to notice.

To further demonstrate the effectiveness of our algorithm, I have added examples of atmospheric optical phenomena other than halos: third and fourth order rainbows and pollen corona.

**Line 182:**
**6.2 Other Atmospheric Optical Phenomena**
The sky-color regression algorithm can also be used efficiently to enhance other colored atmospheric optical phenomena. Fig.13(a) shows an example of quite faint third and fourth order rainbows. The algorithm extracts colored bows from the background sky without clouds. Fig.13(b) shows the diffraction pattern of Japanese cedar pollen corona. The algorithm also works on the sky color near sunset.

[Figure]

**Figure 13:** Applying sky-color regression for images of other atmospheric optical phenomena: (a) third and fourth order rainbows, (b) pollen corona (Japanese ceder).

---

## Author Response (AR3)

**Response to the Editor**

**Revealing Halos Concealed by Cirrus Clouds**

Yuji Ayatsuka

I would like to thank the editor to find errors to correct in our manuscript. I have fixed/modified them.